# Toward General Object-level Mapping from Sparse Views with 3D Diffusion Priors

**Ziwei Liao**    **Binbin Xu**    **Steven L. Waslander**
Institute for Aerospace Studies & Robotics Institute
University of Toronto
{ziwei.liao, binbin.xu, steven.waslander}@robotics.utias.utoronto.ca

**Abstract:** Object-level mapping builds a 3D map of objects in a scene with detailed shapes and poses from multi-view sensor observations. Conventional methods struggle to build complete shapes and estimate accurate poses due to partial occlusions and sensor noise. They require dense observations to cover all objects, which is challenging to achieve in robotics trajectories. Recent work introduces generative shape priors for object-level mapping from sparse views, but is limited to single-category objects. In this work, we propose a **G**eneral **O**bject-level **M**apping system, *GOM*, which leverages a 3D diffusion model as shape prior with multi-category support and outputs Neural Radiance Fields (NeRFs) for both texture and geometry for all objects in a scene. GOM includes an effective formulation to guide a pre-trained diffusion model with extra nonlinear constraints from sensor measurements without finetuning. We also develop a probabilistic optimization formulation to fuse multi-view sensor observations and diffusion priors for joint 3D object pose and shape estimation. GOM demonstrates superior multi-category mapping performance from sparse views, and achieves more accurate mapping results compared to state-of-the-art methods on the real-world benchmarks. We will release our code: https://github.com/TRAILab/GeneralObjectMapping.

**Keywords:** Mapping, Objects Reconstruction, Pose Estimation, Diffusion

## 1 Introduction

Object-level mapping [1, 2, 3, 4, 5, 6, 7, 8, 9] builds a 3D map of multiple object instances in a scene, which is critical for scene understanding [10] and has various applications in robotic manipulation [11], semantic navigation [12, 13] and long-term dynamic map maintenance [14]. It addresses two closely coupled tasks: 3D shape reconstruction [15, 16] and pose estimation [17]. Conventional methods [18, 19, 20] approach these tasks from a perspective of state estimation [21], solving an inverse problem where low-dimensional observations (RGB and Depth images) are used to recover high-dimensional unknown variables (3D poses and shapes) through a known observation process (e.g., projection, and differentiable rendering). However, these methods require dense observations (e.g., hundreds of views for NeRF [18]) to fully constrain the problem. In robotics or AR applications, obtaining such dense observations is challenging due to limitations in the robot's or user's observation angle and occlusions in clustered scenarios. Therefore, it is crucial to develop methods that can map from sparse (fewer than 10) or even single observations.

Human vision can infer complete 3D objects from images despite occlusions by using *prior knowledge* of the objects, which represents the *prior distributions* of the shapes of specific categories, such as chairs, based on thousands of instances observed in daily life. We aim to introduce generative models [22] as providers of *prior knowledge* to constrain the 3D object mapping. Generative models have demonstrated impressive abilities to generate high-quality multi-modal data by learning distributions in large-scale datasets, including texts [23], images [24], videos [25], and 3D mod-

8th Conference on Robot Learning (CoRL 2024), Munich, Germany.

els [26, 27, 28, 29]. Previously, researchers have explored methods, to introduce generative priors to constrain inverse problems in the image field, such as restoration [30] and controllable generation [31]. However, applying generative models to 3D inverse problems remains an open problem.

Two types of generative priors have been explored in the context of 3D inverse problem. (1) **Image Generation Models as 2D Priors**. Researchers [32, 33] use image generation models [24] to generate multi-view images to optimize NeRF parameters for shape reconstruction. However, these models suffer from poor 3D consistency (e.g., multi-face problem [32]) as it is difficult to preserve 3D consistency and identity in image generation models. Additionally, they require dense observations, which limit computation efficiency (e.g., taking hours per object). Achieving high-quality results often necessitates complex modifications, such as retraining the generation model [34, 35, 36]. Furthermore, these methods typically focus on shape generation from texts, rather than on reconstruction *and pose estimation* from multi-view real observations. (2) **3D Shape Generation Models as 3D Priors**. Recent object-level mapping systems [3, 4] directly inject 3D priors from a 3D generation model like DeepSDF [27] by jointly optimizing pose and shape (represented in a learned latent space). Compared to 2D priors, which provide only partial and low-dimensional shape information, 3D priors achieve better 3D consistency and completeness. However, limited by the capacity of an autodecoder model [27], they support only single-category objects, and model only the geometries of shapes without textures, limiting the generalization ability.

We aim to propose an object-level mapping system that introduces valuable generative priors for diverse objects in any scene. Our system is therefore designed to be: (1) *General*: It supports multi-category objects by using a pre-trained generative model as shape prior; (2) *Flexible*: It supports multi-view, multi-modality observation signals (e.g., RGB, depth images and pointclouds), without the need for finetuning the generative model. Retraining large-scale 3D generative models is a process that is both time-consuming and resource-intensive. The attributes of being *General* and *Flexible* are essential when integrating into a complex robotics system for downstream tasks.

Recently, diffusion-based 3D generative models trained on millions of 3D objects and supporting multiple object categories have been released [37, 29]. Additionally, new public large-scale 3D object datasets [38, 39] are becoming available. However, employing diffusion-based models to solve inverse problems remains theoretically unsolved, since diffusion models use score functions to model distributions [24] and cannot directly output probability densities like the VAE-variant models [27]. In this paper, we make a key theoretical contribution by proposing an effective formulation to fuse 3D diffusion priors with 3D sensor observations. As far as we know, we are the first to explore 3D object-level mapping with a multi-category 3D diffusion model for both pose and shape estimation. Our contributions are summarized as follows:

- We propose a 3D object-level mapping system from sparse RGB-D observations, leveraging a pre-trained diffusion-based 3D generation model as a multi-category shape prior;

- We propose an effective optimization formulation that jointly fuses information from multi-view sensor observations and diffusion priors;

- We demonstrated our superior multi-category mapping performance over the state-of-the-art baselines through extensive experiments on the real ScanNet dataset.

## 2 Related Work

### 2.1 3D Object-level Mapping

3D scene understanding is crucial for robots to perform high-level object-oriented tasks such as navigation and manipulation [10]. Early methods in the field of SLAM and SfM [40] built 3D scene models from dense multi-view observations, leveraging multi-view geometric consistency [41]. However, they used a single model to represent both foreground objects and backgrounds, unaware of separate instances [16, 42, 43]. Object-level mapping addresses the problem of estimating objects' shapes and poses as independent instances. SLAM++ [1] was pioneering in this area but requires a

database of CAD models and could not handle unseen objects outside this database. To generalize to unseen objects, researchers have used compact object representation such as quadrics [6, 7] or cuboids [5], but these approaches lack detailed shape information and texture. Methods [44, 2, 45] produce dense representations, such as Signed Distance Function (SDF) fields, and Neural Radiance Fields (NeRFs) [46]. However, they require dense observations to constrain the high-dimensional shape variables, and struggle to achieve complete and accurate results in the presence of occlusions and noise. Differing from the methods above, we develop our method to address object-level mapping from *sparse* observations, which is more relevant for robotic applications.

## 2.2 Generative Models as Shape Priors

Introducing generative models as prior constraints for 3D object pose and shape estimation is an open problem. Currently, the literature explores two kinds of priors. (1) Prior for Specific Single-category. Early 3D generative models [27, 28, 47] train separate models for single-category objects. Object-level mapping systems [8, 3, 4, 48, 9] introduce DeepSDF-like [27] generative models as priors to constrain object shapes. They estimate complete shapes and poses from sparse observations under occlusions, however, limited by the shortcomings of these generative priors, they are limited to a single category. (2) General Prior for Multi-categories. Recent research, including DreamFusion [32] and Magic3D [33], optimizes 3D shapes represented by NeRF through multi-view images from an image generation model [49]. However, since the prior is inherently 2D, it has limited 3D consistency, leading to multi-face artifacts and identity-preserving problems [32]. Although they can generate multi-category 3D shapes from texts, they can not reconstruct from real images or estimate object poses to form a map with multiple instances. In contrast to the methods above, we aim to leverage a 3D prior model with multi-category ability for improved 3D consistency. Recently, diffusion-based 3D generative models [37, 29] have become available, such as Shap-E [29] which is trained on millions of objects in thousands of categories. However, unlike VAE-variant models [27], diffusion-based models are not straightforward to integrate into an object-level mapping system. We discuss this further in the next section.

## 2.3 Pretrained Diffusion Models as Optimization Constraints

Training a diffusion model on millions of 3D objects is time-consuming [37, 29]. We aim to leverage a pre-trained diffusion model without finetuning. One method to control a pre-trained diffusion model is through conditional generation. It introduces extra constraints to guide the original diffusion process, and has been investigated in image generation [31], restoration [30], and 3D human pose generation [50]. However, a conditional generation formulation is not flexible enough to estimate extra variables (i.e. poses) beyond the variable modeled by the diffusion model (i.e. shapes). Another method is to leverage the diffusion model as a constraint in optimization, which is flexible to introduce extra variables and constraints. However, unlike VAE-variant methods that have explicit latent spaces [27, 3, 9], the latent space of diffusion models is still under investigation [51]. Diffusion models output scores instead of densities and require a complex sampling process to generate a sample [24], making it nontrivial to formulate a joint optimization with additional variables. Recently, some work uses optimization with diffusions, in the fields of images [51, 52], 3D human poses [53, 54], and medical CT reconstruction [55]. Most similar to ours, Yang et al. [56] optimized a NeRF with a *single-category* diffusion model, but did not include pose estimation. Differing from the work above, we are the first to use a pre-trained multi-categories diffusion model in an optimization framework for both 3D object pose and shape estimation.

# 3 Methods

## 3.1 Framework Overview

Our system, visualized in Figure 1, estimates 3D object shapes and poses in a scene using two information sources: multiple RGB-D observations and a generative shape prior model. Following

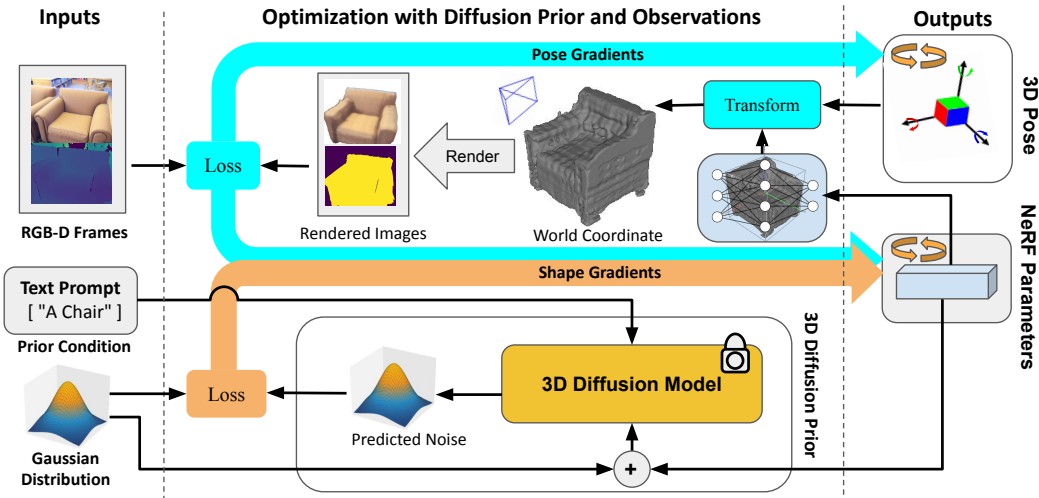

Figure 1: Framework overview. We propose an object-level mapping framework that fuses both multi-view observations and a pre-trained diffusion shape prior model. It generalizes to multi-categories objects, and multiple multi-modalities observations without the need of fine-tuning.

Shap-E [29], object shapes are represented as NeRFs with SDFs, modeling both the geometry and texture, which can be rendered into images or generate meshes.

## 3.2 3D Objects Shapes and Poses Representations

We aim to build a map of the 3D objects $\mathcal{O}$ in a scene. For each object, we estimate its 3D model (a NeRF), $\boldsymbol{Q}_o$, in the object canonical coordinate frame, $\mathcal{F}_{\rightarrow O}$, and a relative pose, $\mathbf{T}_{OW}$, from the world coordinate frame, $\mathcal{F}_{\rightarrow W}$, to the object coordinate frame $\mathcal{F}_{\rightarrow O}$.

**Pose Representation**. A pose transformation with scale, $\mathbf{T} \in \mathbb{R}^{4 \times 4}$, is constructed from a translation vector, $\mathbf{t} \in \mathbb{R}^3$, a rotation vector, $\phi \in \mathfrak{so}(3)$, and a scaling vector, $\mathbf{s} \in \mathbb{R}^3$:

$$\mathbf{T} = \begin{bmatrix} \exp\left(\phi^\wedge\right) & \mathbf{t} \\ \mathbf{0}^T & 1 \end{bmatrix} \cdot \begin{bmatrix} \mathrm{diag}\left(\mathbf{s}\right) & \mathbf{0} \\ \mathbf{0}^T & 1 \end{bmatrix} \tag{1}$$

where $\exp\left(\cdot\right)$ is the exponential mapping from the Lie Algebra to the corresponding Lie Group. The operator $\left(\cdot\right)^\wedge$ converts a vector to a skew-symmetric matrix. Thus, we can represent the pose with a 9-DoF vector: $\xi = [\mathbf{t}, \phi, \mathbf{s}] \in \mathbb{R}^9$. As a mapping system, we assume camera poses, $\mathbf{T}_{WC} \in SE(3)$, are known, e.g., by off-the-shelf SLAM methods [43], or by robot kinematics [57].

**Shape Representation**. We parameterize the object's shape with a Neural Radiance Field (NeRF) [18] through a neural network $f_\Theta(.)$ with weights $\Theta$. For any given 3D point, it outputs its density, $\sigma$, color, $\mathbf{c}$, and illumination, $i$, and can be rendered into RGB and depth images. Following Shap-E [29], it further outputs SDF values, $s$, and can generate meshes via Marching Cubes:

$$\sigma, \mathbf{c}, i, s = f_\Theta(\mathbf{x}, \mathbf{d}) \tag{2}$$

where $\mathbf{x} \in R^3$ is the given 3D coordinate and $\mathbf{d} \in R^3$ is the viewing direction.

## 3.3 Leveraging Generative Models as Shape Priors

Common 3D objects, such as chairs and tables, exhibit diverse shapes and textures. This translates to high-dimensional NeRF parameters $\Theta$. To further constrain the process, we propose to leverage a generative model with learnable parameters $\beta$, denoted by $g_\beta(.)$, to learn an approximate posterior distribution $\hat{P}(\Theta|C)$ of the true posterior distribution $P(\Theta|C)$. Here $C$ represents the conditioning information, e.g., an image or a text prompt. The generative model $g_\beta(.)$ takes this conditioning information, $C$, as input and outputs a *conditional* distribution of the NeRF parameters $\Theta$:

$$\hat{P}(\Theta|C) = g_\beta(C) \tag{3}$$

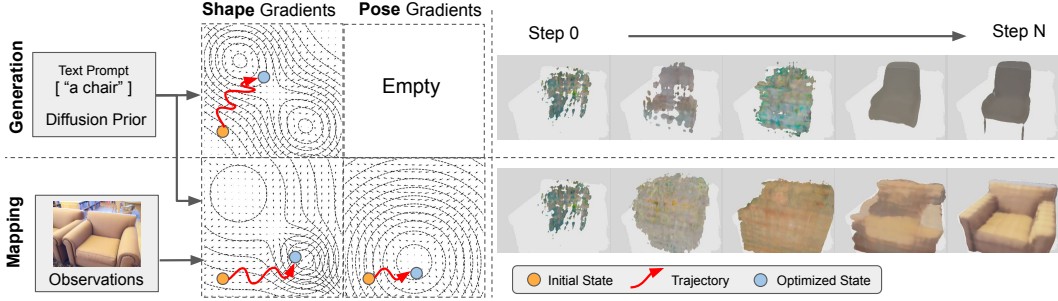

Figure 2: An illustration of the gradient fields and optimization process. The gradient fields from two sources, a diffusion prior originally for *Generation*, and multi-view non-linear observation constraints, are effectively fused into a shape and pose optimization formulation for *Mapping*.

**Diffusion Models**. We leverage Shap-E [29], a pre-trained diffusion model as $g_\beta(.)$, which has been trained on millions of 3D objects. As shown in Figure 2, to generate a sample $\Theta_i \in \hat{P}(\Theta|C)$, we start from a random noise $\Theta_T$ and refine it through a series of diffusion steps $\epsilon_\beta(.)$ for timestamp $t = T, ..., 1$, according to a noise schedule. This process produces the final noise-free sample $\Theta_0$:

$$\Theta_{t-1} = \epsilon_\beta(\Theta_t, C, t) \tag{4}$$

### 3.4 Optimization with Diffusion and Extra Observations

To update $\Theta$ with multi-view RGB-D observations and further estimate a new unknown pose variable $\mathbf{T}$, we propose an optimization formulation that integrates additional observations into the pre-trained diffusion model, as shown in Figure 2. We refer $\mathbf{T}$ as $\mathbf{T}_{OW}$ without further notice. Given $M$ observation frames $\{F_i\}_{i=1}^M$ and a condition $C$, we aim to estimate a Maximum Likelihood Estimation for the unknown variables pose $\mathbf{T}$ and shape $\Theta$, from a joint distribution of $P(\mathbf{T}, \Theta|F_1, \ldots, F_M, C)$. By derivation (Proof provided in Appendix), we get a convenient form for numerical optimization:

$$\hat{\mathbf{T}}, \hat{\Theta} = \arg\max_{\mathbf{T}, \Theta} \sum_i \underbrace{\log P(F_i|\mathbf{T}, \Theta)}_{\text{Observations}} + \underbrace{\log P(\Theta|C)}_{\text{Diffusion Priors}} \tag{5}$$

**Diffusion Priors**. The term $\log P(\Theta|C)$ measures the likelihood of current $\Theta$ under a condition $C$ by the diffusion model. However, diffusion models can not directly output the probability density [24]. We derive a method to calculate gradients from diffusion priors for optimization, following a variational sampler [58], originally designed for controlling image generation. A diffusion model is trained to predict the noise $\epsilon$ in a noisy variable $\Theta_t$ at timestamp $t$, under a condition $C$:

$$\min_\beta = \mathbb{E}_{t,\epsilon} \|\epsilon_\beta(\Theta_t, C, t) - \epsilon\|_2^2, \quad t \in \mathcal{U}(0, T), \quad \epsilon \in \mathcal{N}(0, I_n) \tag{6}$$

We fix the weights $\beta$ of the pre-trained diffusion model, and use it to get the gradients $\Delta_\Theta \log P(\Theta|C)$ of the current shape $\Theta$. First, we add noise to the current shape parameter $\Theta$ to get a noisy $\Theta_t$ with the predefined noise schedule of the diffusion model at timestamp $t$:

$$\Theta_t = \alpha_t \Theta + \sigma_t \epsilon, \quad \epsilon \in \mathcal{N}(0, I_n) \tag{7}$$

where $\sigma_t$ and $\alpha_t$ are constants of the schedule, and $\epsilon$ is a randomly sampled noise. Then, we predict the added noise and evaluate the predicted error $\Delta_\beta(\Theta, C, t)$ with the diffusion model:

$$\Delta_\beta(\Theta, C, t) = \epsilon_\beta(\Theta_t, C, t) - \epsilon \tag{8}$$
$$= \epsilon_\beta(\alpha_t \Theta + \sigma_t \epsilon, C, t) - \epsilon, \quad \epsilon \in \mathcal{N}(0, I_n) \tag{9}$$

To eliminate the timestamps $t$, we randomly sample $K$ times from a uniform distribution $\mathcal{U}(0, T)$ and take an expectation:

$$\Delta_\beta(\Theta, C) = \frac{1}{K} \sum_{i=1}^K \Delta_\beta(\Theta, C, t_i), \quad t_i \in \mathcal{U}(0, T) \tag{10}$$

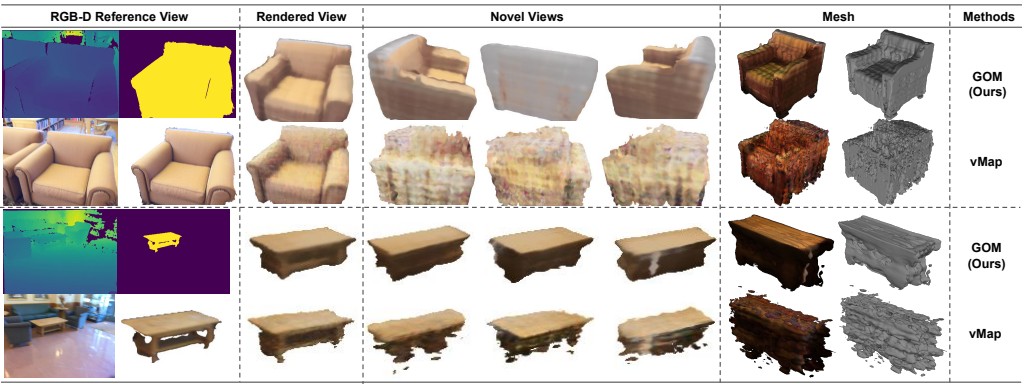

| RGB-D Reference View | Rendered View | Novel Views | Mesh | Methods |
|---|---|---|---|---|

Figure 3: Prior Effectiveness: Using priors, GOM (Ours) can render higher-quality 3D consistent views and generate 3D meshes with fewer artifacts compared to vMap. Results are based on 10 RGB-D views.

As illustrated in Figure 2, the predicted noise of a diffusion model naturally models the gradient field [24] of the prior distribution. Thus, we take the predicted error $\Delta_\beta(\Theta, C)$ as the gradient to update $\Theta$ under a prior condition $C$:

$$\nabla_\Theta \log P(\Theta|C) = \Delta_\beta(\Theta, C) \tag{11}$$

We also present ablations with other alternatives in the Appendix.

**Observations**. The term $\sum \log P(F_i|\mathbf{T}, \Theta)$ measures the geometric and texture consistency from multi-view observations. For each observation $F_i = \{I_i, D_i, \mathbf{P}_{WC_i}\}$, we use differentiable rendering [18] to render an RGB image $\hat{I}_i(\mathbf{T}_{OW}, \Theta, \mathbf{P}_{WC_i})$ and a depth image $\hat{D}_i(\mathbf{T}_{OW}, \Theta, \mathbf{P}_{WC_i})$ with the camera projection matrix $\mathbf{P}_{WC_i}$. We then formulate an L2 loss:

$$\log P(F_i|\mathbf{T}, \Theta) \propto |\hat{I}_i - I_i|^2 + |\hat{D}_i - D_i|^2 \tag{12}$$

This term can propagate gradients to update both poses $\mathbf{T}_{OW}$ and shapes $\Theta$ as shown in Figure 2.

**Iterative Update**. We first coarsely initialize the unknown variables shape $\Theta_0$ and pose $\mathbf{T}_0$. Then, we iteratively refine for $J$ steps considering a prior condition $C$ and observations $\{F_i\}$:

$$\Theta_j, \mathbf{T}_j = Refine(\Theta_{j-1}, \mathbf{T}_{j-1}, C, \{F_i\}), \quad \text{for} \quad j = 1, ..., J \tag{13}$$

The $Refine()$ process has two steps. First, a *Prior Step* to update $\Theta$ with diffusion using Eq. 11:

$$\Theta_j^* = \Theta_{j-1} + \lambda_p \nabla_\Theta \log P(\Theta|C)|_{\Theta=\Theta_{j-1}} = \Theta_{j-1} + \lambda_p \Delta_\beta(\Theta_{j-1}, C) \tag{14}$$

where $\Theta_j^*$ is a middle shape fused with prior. Then, an *Observation Step* to update both $\Theta$ and $\mathbf{T}$:

$$\Theta_j = \Theta_j^* + \lambda_o \sum_i \nabla_\Theta log P(F_i|\mathbf{T}, \Theta)|_{\Theta=\Theta_j^*, \mathbf{T}=\mathbf{T}_{j-1}} \tag{15}$$

$$\mathbf{T}_j = \mathbf{T}_{j-1} + \lambda_o \sum_i \nabla_\mathbf{T} \log P(F_i|\mathbf{T}, \Theta)|_{\Theta=\Theta_j^*, \mathbf{T}=\mathbf{T}_{j-1}} \tag{16}$$

where $\lambda_p$ and $\lambda_o$ are hyperparameters to balance the influence of priors and observations. Finally, we obtain the refined $\mathbf{T}_J$ and $\Theta_J$ after $J$ steps optimization.

## 4 Experiments

**Datasets**. We conducted experiments utilizing the ScanNet [59] dataset, which comprises RGB-D scans of real indoor scenes. It presents significant challenges to object mapping methods due to real-world occlusions, clusters, blurs, and sensor noise. We use ground-truth shape annotations of ShapeNet [60] meshes provided by Scan2CAD [61] for evaluation. We focus on objects that have a

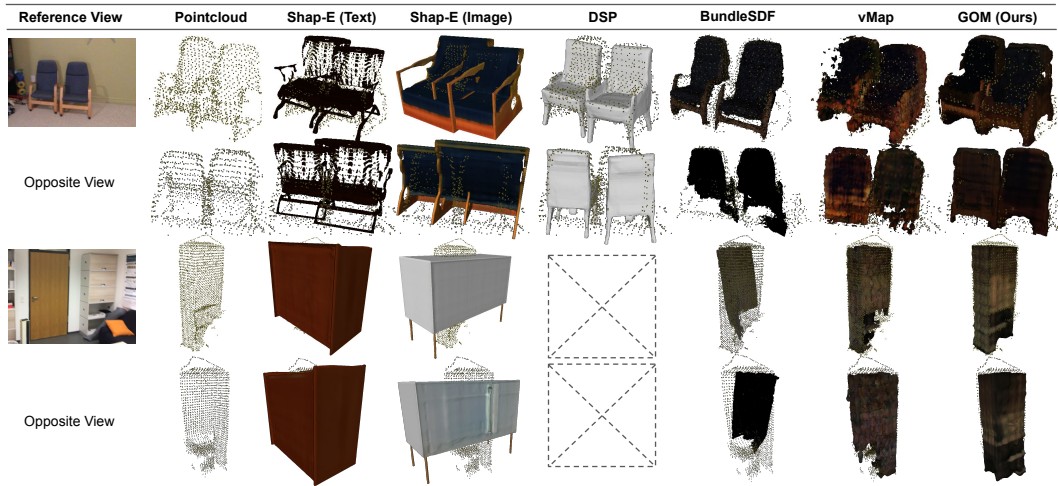

| Reference View | Pointcloud | Shap-E (Text) | Shap-E (Image) | DSP | BundleSDF | vMap | GOM (Ours) |

Figure 4: Mapping Performance: GOM (Ours) outputs 3D object shapes and poses that align with the inputs and further completes the occluded parts. GOM can also generalize to multiple categories. Results are based on 10 RGB-D views.

minimum of 10 available views, where at least 50% of the object is visible in each view. This results in a total of 206 scenes, encompassing 1036 object instances across 7 categories. These categories include chairs, tables, cabinets, sofas, bookshelves, beds, and bathtubs.

**Baselines and Metrics**. We compared our method against the following state-of-art RGB-D object-level mapping methods: (1) *Shap-E+ICP*, which uses Shap-E [29] to generate a NeRF model conditioned on an input image, and then matches RGB-D inputs with ICP [62]; (2) *DSP* [3], which optimizes objects shapes and poses with DeepSDF [27] as shape priors; (3) *vMap* [45], which optimizes an independent NeRF [18] from scratch with a fixed pose for each 3D object; (4) *BundleSDF* [20], which optimizes objects' NeRF [18] and poses for tracking. To ensure a fair comparison, we utilize the ground truth camera poses, segmentation masks, and data association provided by the dataset across all systems. For pose accuracy, we use Intersection over Union (IoU) to evaluate the bounding boxes of the estimated and ground truth shapes in world coordinates. For shape accuracy, we calculate Chamfer Distance (CD) after transforming objects into world coordinates.

**Implementation Details**. We optimize for 200 steps, with a fixed learning rate of 0.5. For every 2 steps, we diffuse once with a timestamp randomly sampled from the Shap-E diffusion schedule. We use the text prompt "a {category}" as the condition for Shap-E, where the category comes from the segmentation mask. We randomly initialize the shape variable from a Gaussian distribution and a coarse pose using ICP matching between the input depth point cloud and an average shape of the corresponding category, generated from Shap-E with the prompt "a {category}". The same ICP pose initialization method is applied for DSP and vMap to ensure a fair comparison. To test the potential performance of vMap [45], we replace its iMAP representation with NeRF [18] and run more optimization iterations until it converges. We provide more details in the Appendix.

### 4.1 Multi-view Mapping Performance

We present the mapping performance with 1, 3, and 10 RGB-D views on chairs in Table 1. We show qualitative results in Figure 3 and Figure 4. Compared to vMap [45], which lacks priors as additional constraints, our approach yields significant improvements. This is particularly evident with fewer views, where observations are compromised by occlusions and noise. In these instances, prior information serves as a valuable additional constraint. Compared to DSP [3], our method produces more detailed shapes with textures, and outperforms it in both 3 and 10 views. DSP achieves better CD in single-view cases due to a smaller latent space with stronger constraints specifically trained on Chairs. However, it can not generalize to other categories. Shap-E can generate reason-

| Views | 1 | | 3 | | 10 | |
|---|---|---|---|---|---|---|
| Methods | IoU ↑ | CD ↓ | IoU ↑ | CD ↓ | IoU ↑ | CD ↓ |
| Shap-E [29] + ICP | 0.337 | **0.169** | - | - | - | - |
| DSP [3] | - | 0.202 | - | 0.173 | - | 0.171 |
| vMap [45] | 0.370 | 0.282 | 0.384 | 0.184 | 0.413 | 0.158 |
| GOM (Ours) | **0.401** | 0.267 | **0.409** | **0.166** | **0.429** | **0.157** |

Table 1: Mapping performance from 1, 3, and 10 RGB-D views on ScanNet Chairs dataset. Since Shap-E can only take in one image, we present a single-view result for Shap-E+ICP.

| Tasks | Methods | Chairs | Tables | Cabinets | Sofas | Bookshelfs | Beds | Bathtubs | **Average** |
|---|---|---|---|---|---|---|---|---|---|
| Mapping | Shap-E [29] + ICP * | 0.169 | 0.430 | 0.268 | 0.162 | 0.392 | 0.813 | 1.91 | 0.437 |
| | DSP [3] | 0.171 | - | - | - | - | - | - | - |
| | vMap [45] | 0.158 | 0.249 | 0.200 | 0.220 | 0.315 | 0.729 | 1.53 | 0.350 |
| | GOM (Ours) | **0.157** | **0.173** | **0.194** | **0.136** | **0.258** | **0.500** | **1.32** | **0.290** |
| Recon | Shap-E [29] Image * | 0.206 | 0.184 | 0.101 | 0.186 | **0.084** | 0.266 | 0.139 | 0.188 |
| | Shap-E [29] Text | 0.211 | 0.189 | 0.260 | 0.187 | 0.143 | 0.362 | 0.207 | 0.211 |
| | GOM (Ours) | **0.102** | **0.119** | **0.096** | **0.137** | 0.098 | **0.132** | **0.110** | **0.107** |

Table 2: Mapping performance on 7 object categories on ScanNet dataset from 10 RGB-D views for DSP, vMap, and GOM (Ours). * Since Shap-E can only take in one image, we present a single-view result for Shap-E+ICP and Shap-E Image for reference. Metrics: Chamfer Distance (CD).

able chairs from image conditions, as shown in Figure 4. When combined with ICP, it exhibits a low CD, but struggles to faithfully reconstruct objects. Its performance decreases significantly for large objects such as tables, beds and bathtubs, as shown in Table 2. GOM (Ours) achieves better IoU and can continue to improve when provided additional observations. We qualitatively compare to BundleSDF [20] in Figure 4 and we see that GOM delivers better mapping results as it better completes the unseen parts of objects with prior information.

## 4.2 Multi-categories Performance

We benchmark the open-vocabulary mapping performance for objects across 7 categories using 10 RGB-D views in Table 2. Our approach outperforms the naive use of ICP matching with shapes generated from Shap-E. The ICP matching stuck to a local minimum for the pose with a fixed shape. DSP [3] is a single-category system that requires separate network weights for each category, and since only the chair model is officially available, we present its performance solely for this category. This limitation also restricts the system's applicability. Compared to vMap [45], our approach yields an improved average CD and enhancements across most categories. This highlights the efficacy of our optimization formulation in integrating prior information with observations. We further demonstrate the *Reconstruction* performance in Table 2 (Recon), given the ground truth object poses. Our performance significantly improves compared with Mapping, demonstrating the potential for enhanced performance when more accurate initial poses are available. Our approach outperforms both the text and image-conditioned Shap-E model. This underscores that additional observations can enhance a pre-trained generative model for reconstruction tasks. *We present more experiments on the CO3D dataset and analysis in the Supplementary Materials.*

## 5  Conclusion

We present a General Object-level Mapping system, *GOM*, leveraging a pre-trained diffusion-based 3D generation model as shape priors. We propose an optimization formulation to couple multi-view RGB-D observations, and diffusion priors to constrain shapes and poses for 3D objects. We achieve state-of-the-art mapping performances among multiple categories without further finetuning. Further exploring the diffusion shape priors into inverse problems with more constraints, e.g., temporal constraints for dynamic tracking, and spatial constraints for complete SLAM, and the application to downstream robotics tasks such as robotics manipulation, will be valuable future work directions.

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

# 6 Appendix

## 6.1 Experiments on CO3D dataset

To demonstrate the generalization ability across diverse categories, we conducted additional experiments on 10 categories from the CO3D dataset [63]: Toy Truck, Bench, Donut, Broccoli, Toy Train, Apple, Teddy Bear, Hydrant, Book, and Toaster. We compared our results with the baseline method vMap [45], as shown in Figure 5. Our approach consistently outperformed the baseline, benefiting from the incorporation of generative priors from a single pre-trained model.

## 6.2 More Qualitative Visualizations

**Visualization on More Categories on ScanNet dataset**. We further illustrate the effectiveness of our approach in generalizing to a variety of categories through qualitative visualizations, as shown in Figure 6. Unlike the DSP method [3], which is based on a single-category model, DeepSDF [27], our system is capable of supporting multiple object categories using a single pre-trained diffusion model, Shap-E [29].

**Scene-level Visualization**. Figure 7 showcases a 3D map visualization of a scene on the ScanNet dataset, containing multiple objects from various categories, including four chairs, a sofa, and a table. Each instance is independently reconstructed using 10 RGB-D views.

**Visualization of Multi-view Inputs**. We visualized 10 input images, 3D camera poses, and corresponding object meshes in Figure 8, which includes two objects from the ScanNet dataset and one object from the CO3D dataset. The ScanNet dataset presents a challenging camera trajectory, with all inputs gathered from a single direction relative to the objects. This scenario is typical in real-world applications such as robotics and augmented reality, where a robot or user cannot easily circle an object. It also underscores the importance of sparse view mapping, where prior information is crucial for completing and providing reasonable estimates for occluded parts. An extreme case is illustrated where all camera views have small baselines, resembling single-view mapping. In the CO3D dataset, the cameras are intentionally positioned to circle the objects; however, we randomly sampled only ten views from this trajectory, resulting in a sparse coverage of the objects. We demonstrate that our system can effectively handle all these situations.

## 6.3 More System Evaluations and Discussions

**Performance on Single-view Inputs**. Single-view mapping is particularly challenging due to its highly ill-posed nature. All systems experience significant performance losses when relying on only a single view. Our main challenges stem from the high-dimensional shape representation required for NeRF, which models multiple categories of objects, encompassing both texture and shape. In contrast, DSP employs DeepSDF, an SDF-based representation specifically trained on a single category, focusing solely on shapes. This approach has a much smaller parameter space and requires fewer constraints. However, this limited parameter space restricts its ability to generalize to other categories. Our method, on the other hand, can generalize across multiple categories using a single model and can continue to improve with additional observations. With the assistance of prior knowledge, we achieve effective enhancements in single-view scenarios compared to the baseline method, vMap. Additionally, we have a parameter that allows us to increase the weight of prior constraints during optimization, enabling us to place greater trust in the prior information for producing complete and reasonable mapping results.

**Performance on Out-of-distribution Objects**. Dealing with out-of-distribution objects is a critical challenge for developing a general mapping system. In this paper, we present a formulation that leverages prior knowledge from a pre-trained generative model. While we use Shap-E as a representative example at the time of writing, it's important to note that our method is not specifically tailored to Shap-E but applies to a broader class of diffusion models. The field of 3D generation

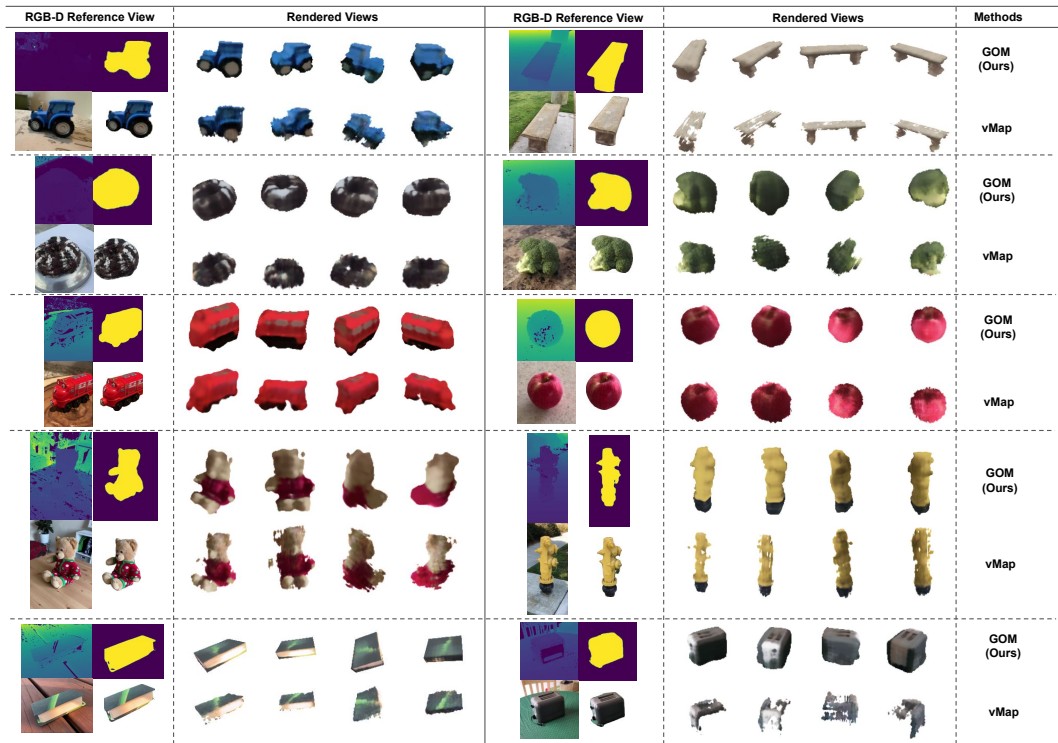

Figure 5: Effectiveness of Priors Across 10 Categories on the CO3D Dataset: Toy Truck, Bench, Donut, Broccoli, Toy Train, Apple, Teddy Bear, Hydrant, Book, Toaster, compared to vMap. The results are based on 10 RGB-D views.

models is rapidly evolving, with larger datasets (e.g., ObjaverseXL [39], OmniObject3D [64]) and more powerful generative models emerging.

Our proposed formulation allows for seamless integration of more advanced models without requiring any fine-tuning, enabling a plug-and-play approach. We demonstrate that the current Shap-E model is effective for a diverse range of objects, as shown by our results on both ScanNet and CO3D. While generative models may not provide detailed structural constraints for out-of-distribution objects, they still offer valuable general priors, such as completeness and smoothness. This is illustrated in the experiment in Figure 9, where a prompt like "an object" yields useful results.

Our system effectively combines observations and prior constraints to identify an optimal solution from both sources. When the prior information is less accurate, we can adaptively adjust the weights to rely more on observations for out-of-distribution objects, tailored to specific applications. Furthermore, it would be valuable future work to quantify the level of out-of-distribution characteristics, such as uncertainty [9, 65], and to self-adjust the weights to enhance mapping performance.

**Text Prompts and Segmentation Methods**. When selecting text prompts as prior conditions, our system demonstrates certain flexibility regarding the content of the text. In our experiments, we provide an example using the prompt "a category," where category can be easily obtained from off-the-shelf object detection or segmentation algorithms. We also conducted a new experiment to analyze the sensitivity to the quality of the text prompt, as shown in Figure 9. In this qualitative analysis, we focused on a chair that was only observed from the front, using three types of prompts: (1) Precise label: "a chair"; (2) General label: "an object"; and (3) Two incorrect labels: "a table" and "a ball." Our findings indicate that our system can robustly leverage prior knowledge from varying levels of label specificity by optimizing both the observations and the prior information to converge on a solution that integrates both aspects. Particularly for occluded areas with insufficient observational data, the prior knowledge effectively constrains the reconstruction to produce a reasonable

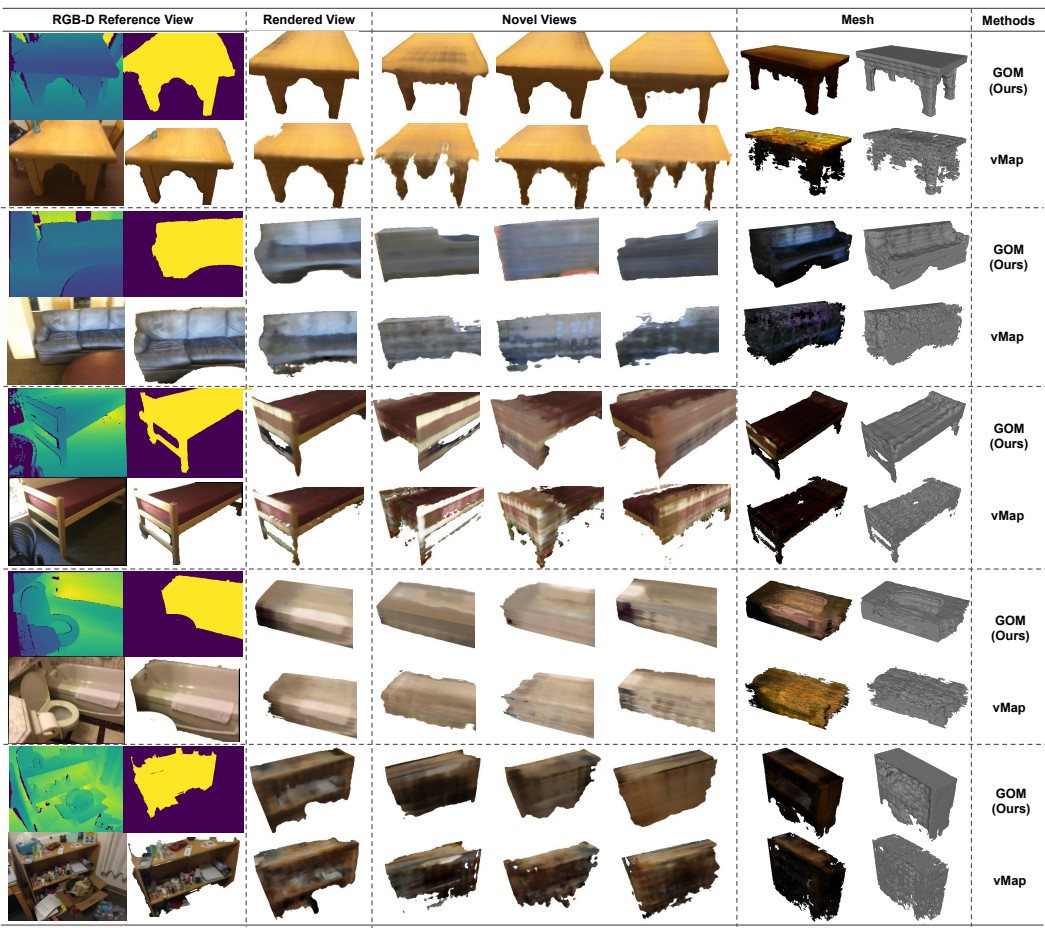

Figure 6: Effectiveness of Priors Across Multiple Categories: Leveraging priors, our method (GOM) can produce higher-quality, 3D-consistent views and generates 3D meshes with fewer artifacts compared to vMap. The results are based on 10 RGB-D views.

and smooth surface compared to vMap without any prior, even when using a general prompt like "an object."

In cases where incorrect labels were used, such as "a table" and "a ball," some artifacts were introduced into the structure (e.g., a ball-like back part of the chair). However, the results were still significantly more coherent than the blurry areas observed in the baseline vMap. For the segmented areas, we utilized the provided "ground truth" masks from the ScanNet dataset, which are not manually annotated but projected from segmented point clouds. These masks may still contain artifacts and missing parts, and our experiments on ScanNet highlight the system's performance under imperfect input conditions. Additionally, we have parameters in place to balance the weight between observation confidence and prior knowledge (as detailed in Eq. 14-16 of the main paper). For future work, it will be valuable to model the uncertainty of the segmentation algorithm to enable adaptive weight balancing for improved robustness.

**Global Map Consistency**. We provide a qualitative scene-level result in Figure 7, demonstrating that our reconstructed shapes from multiple objects are consistent with the overall scene point cloud. In our main experiments, we evaluate each object individually to focus on validating the effectiveness of the generative prior model, which influences each object independently. Our contribution in this paper is distinct from other research efforts aimed at constructing a globally consistent map.

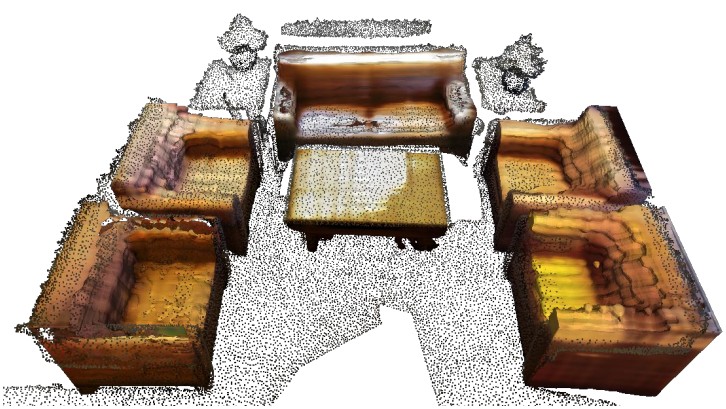

Figure 7: Scene-Level Visualization: An example of a reconstructed 3D map of a scene including four chairs, one sofa, and one table, all constrained with the same prior network.

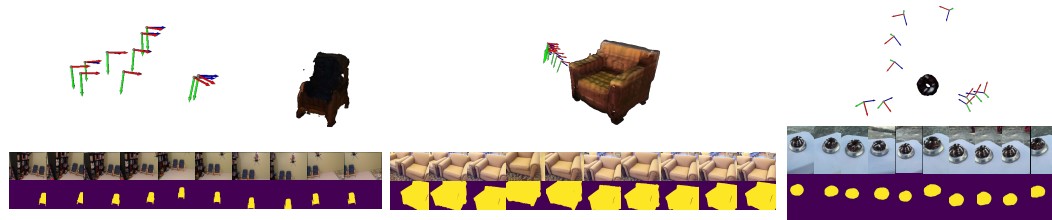

Figure 8: Input images from 10 RGB-D views. Sequences from the ScanNet dataset contain only observations from one side, which is more challenging for the occluded unseen parts. Sequences from CO3D contain 360-degree observations of objects.

| Inputs | | Outputs |
|---|---|---|
| **RGB-D Reference View** | **Text Prompt** | **Rendered Views** |
| | **Precise Label** "A chair" | |
| | **General Label** "An object" | |
| | **Wrong Label 1** "A table" | |
| | **Wrong Label 2** "A ball" | |
| | **vMap** (w/o Prior) | |

Figure 9: Sensitive analysis of the input text prompt, when giving precise, general, and wrong labels for the generative prior model. We find that our system can robustly leverage prior from different information-level labels, by optimizing both information from observations and prior knowledge to find a converged point that can match both parts.

Thanks to our optimization-based formulation, which combines both priors and observations, we can flexibly introduce additional constraints for global consistency. We can leverage well-established techniques from optimization-based SLAM and Structure from Motion (SfM), such as sliding window optimization and loop closure, to enhance global map consistency. We believe our approach can be extended to incorporate further constraints for global consistency, including cross-object semantic relationships and geometric supporting relationships, which can be represented as non-linear

constraints in an optimization framework. However, these enhancements require non-trivial engineering and are currently beyond the scope of this paper.

**Deployment in real applications**. We provide guidance for deploying our proposed method in real-world applications such as robotics to facilitate multi-object mapping. Data association—assigning observations of the same object from multiple frames—is crucial. There are various well-studied methods for data association, including the use of image features (such as manually defined ORB/SIFT or learned features like SuperPoint), ICP matching with point clouds, or probabilistic data association approaches [66, 67]. For camera pose estimation, we can employ offline calibration to determine intrinsic and extrinsic parameters, possibly using additional sensors such as IMUs, or implement online methods like SLAM [43]. We support two types of input for prior conditions: text prompts and segmented images. Both inputs can be easily derived from the input RGB-D images, for example, by utilizing off-the-shelf segmentation algorithms like groundedSAM [68]. Our text prompts are highly flexible; as demonstrated in the experiments in Figure 9, even a general prompt like "an object" can effectively constrain occluded parts. In the context of an autonomous robot, integrating extra text descriptions opens up new avenues for fusing information to construct a map. For instance, in human-robot interaction scenarios, a robot could ask the user, "What is the object in front of me?" and the user might respond, "It's a chair." This interaction provides valuable prior knowledge for the object mapping system. This capability enhances traditional SLAM and mapping systems by allowing them to gather information and constraints beyond mere sensor observations.

**Relative Pose Definition and Evaluation**. In our experiments, the Intersection over Union (IoU) is calculated between the bounding boxes of the estimated and ground truth shapes in world coordinates, rather than between voxelized meshes. This choice means that the metric is not sensitive to the detailed reconstruction quality. Evaluating independent pose errors—such as directly assessing translation, rotation, and scale relative to ground truth poses—does not provide a reasonable metric in our pose formulation. This limitation is shared by other object mapping systems like vMap, DSP, and Shap-E+ICP. In these systems, the pose is defined as a transformation from a canonical object coordinate system to the world coordinate system, which is closely linked to the shapes of objects in that canonical space. However, shape representations are not unique. For example, when using DeepSDF (for DSP) or NeRF representations (for vMap, Shap-E+ICP, and our method), shapes can exhibit different size ratios and orientations (e.g., a 90-degree rotation around the UP axis). Consequently, reconstructing a 3D object in the world involves multiple combinations of estimated poses and shapes in canonical space that can lead to a correct result. Therefore, there is no single, unique pose to evaluate against a "ground truth" pose. It is important to emphasize that, in applications such as robotics, our primary concern is the performance of the final 3D representation in the world, which encompasses both shape and pose, rather than evaluating a relative "pose" alone. Thus, we opt to directly evaluate the final outputs in world coordinates, reporting IoU and Chamfer Distance (CD) as quantitative metrics, accompanied by qualitative images to demonstrate performance relative to the baselines.

## 6.4 Ablation Study

**Methods to Fuse Observations and Diffusion Priors**. We compare three strategies to fuse observations and diffusion priors, as shown in Table 3. (1) *Optimize then diffuse*, which first optimizes shape and pose with geometric loss only for a given number of steps, and then uses the diffusion model to diffuse the shape. We notice that the information from observations is often lost during the post-diffusion process. Consequently, the ultimate shape diverges from the ground truth, resulting in a large metric error. (2) *Diffuse then optimize*, which first uses the diffusion model to generate a shape with a text condition, then uses the geometric loss to optimize both shape and pose. We observe that the unobserved segment of the shape is prone to corruption during the post-optimization process. Ultimately, this leads to a performance level that is similar to optimizing using only geometric observations without priors, which also remains more artifacts in the meshes and renderings. (3) *Jointly Optimize and Diffuse*, which fuses prior and observations iteratively during optimization with both diffusion prior and geometric loss so that both sources of information are active. This com-

| Items | IoU $\uparrow$ | CD $\downarrow$ |
|---|---|---|
| Ours w/ *Optimize then Diffuse* | 0.344 | 0.182 |
| Ours w/ *Diffuse then Optimize* | 0.416 | 0.160 |
| Ours w/ *Jointly Optimize and Diffuse* | **0.429** | **0.157** |
| Gradients - *DirectDiffuse* | 0.338 | 0.222 |
| Gradients - *NoisePredict (Ours)* | **0.429** | **0.157** |
| Ours w/ Image Condition | **0.436** | 0.160 |
| Ours w/ Text Condition | 0.429 | **0.157** |

Table 3: Ablation study on the strategies to fuse both observations and diffusion prior. Results are from 10 RGB-D views on Chairs of ScanNet dataset.

bined optimization can effectively merge constraints from both sources, thereby achieving superior performance compared to the other strategies.

**Methods to Calculate Gradients from a Pre-trained Diffusion Model**. The gradients derived from both the diffusion model and the observations are high-dimensional. Employing a method to effectively combine these gradients to guide the variable toward a convergence point is not a straightforward task. We compare our method with another to demonstrate the effectiveness of our approach, as shown in Table 3. (1) *NoisePredict (Ours)*. In Section 3.4 of the main content, we discuss our method to use the pre-trained diffusion model to predict the added noise and propagate back the error as gradients. This implicitly constrains the shape variable to lie inside the distribution modeled by the diffusion model, where it is trained to accurately predict the added noise. (2) *DirectDiffuse*, which directly uses the diffusion model to predict a less noisy version of the current shape for one step, as $\Theta_{t-1} = \epsilon_\beta(\Theta_t, C, t)$. Yang et al. [56] also use this method to leverage shape prior constraints from a single-category diffusion model for object reconstruction. Our task is more difficult with the extra unknown variable of pose. As shown in Table 3, *DirectDiffuse* underperforms in comparison to *NoisePredict (Ours)*. We attribute this to two primary factors. Firstly, making a pre-trained diffusion model to accurately predict a denoised variable is a challenging task. Secondly, each step of *DirectDiffuse* necessitates a precise timestamp $t$ to denote the level of noise within the current variable, which becomes particularly complex when jointly optimized with gradients from observations. In contrast, our gradients can be derived from randomly sampled, uniformly distributed timestamps. This allows for flexible diffusion across arbitrary steps without the stringent requirement to adhere to the noise schedule from $T$ to 0.

**Input Conditions**. We evaluate both input conditions supported by Shap-E model [29], image and text, as shown in Table 3. Each has its unique strengths and weaknesses, contingent on the specific applications. The image modality, which contains detailed prior information of a specific instance such as texture and shape, is nonetheless limited by the quality of the segmentation task. A corrupted or occluded mask can result in a corrupted 3D shape prior. On the other hand, a simple text prompt like "a chair" can provide a general distribution of complete shapes within the category, albeit without some instance-specific details. This approach allows the details to be constrained by the observations. Future work could explore the use of more complex text prompts and the fusion of multiple multi-modal priors to enhance the effectiveness and accuracy of prior constraints.

## 6.5 Computation Analysis

We conducted an evaluation of the system's computation using 10 RGB-D views on a 16GB V100 GPU. For each instance, GOM (Ours) requires 43.0 seconds for 200 optimization iterations, which includes 100 diffusion steps. In comparison, vMap [45] requires 38.1 seconds to complete 200 optimization iterations, utilizing only geometric constraints. Our method, leveraging diffusion prior, significantly enhances the quality with minimal computational overhead. The Shap-E model [29] requires 45.6 seconds to generate a single instance by diffusing from random noise via a computationally intensive sampling process. Our method, utilizing the prior information stored inside

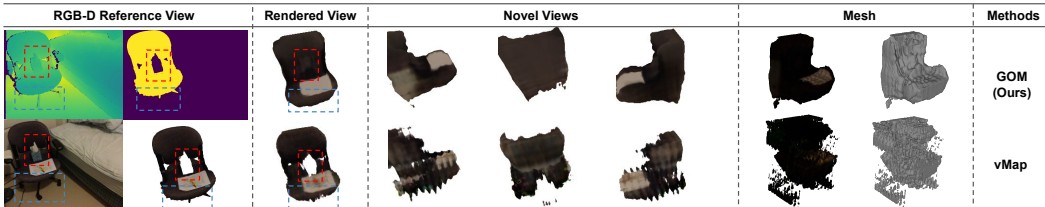

Figure 10: Failure Case: An instance where input observations are occluded and contain corrupted masks. While our method manages to complete part of the object compared to the baseline, it fails to fully complete thin elements such as legs and handles.

Shap-E, can achieve faster reconstruction than the original generation model. DSP [3] requires 32.3 seconds for 200 iterations, a speed benefiting from a smaller latent space provided by DeepSDF [27]. However, it is constrained to a single category and lacks texture information. We adopt a strategy of averaging the total number of rays sampled from multiple frames. Consequently, when more frames are available, the computation time remains nearly identical for 1, 3, and 10 views. BundleSDF [20] reconstructs an object's Signed Distance Function (SDF) and appearance field from scratch, without leveraging any prior information. It utilizes 17 keyframes from the same test scene to reconstruct the object and carries out 2500 iterations to optimize both the neural fields and the object pose. The cumulative running time is 188.6 seconds, partitioned into 118.2 seconds for pose graph optimization and 70.4 seconds for global optimization.

Depending on the specific applications, parameters such as the number of optimization steps, diffusion steps, and sampled rays can be adjusted to balance accuracy and computation. As a direction for future work, the implementation of an incremental mapping framework, as opposed to batch optimization from scratch, could further expedite online applications.

### 6.6 Failure Case and Limitation

The ScanNet dataset presents challenges due to occlusions and sensor noise. We illustrate a representative failure case in Figure 10. When the input observations are severely occluded or contain incomplete masks, our method can partially complete the object (for instance, the center occluded part by the tissue placed on the chair), but it fails to complete thin elements like legs and handles that the mask does not cover. The paper and pen placed on the chair are reconstructed as part of the texture. Despite these challenges, our method still generates a smoother surface with significantly fewer artifacts compared to the baselines. Future improvements could include the use of a more powerful segmentation model, such as SAM [69], and adaptively increasing the weights of the prior in areas with corrupted observations. Further, more flexible shape representation beyond a NeRF, such as Gaussian Splatting [70] can be explored to better model the details of objects.

We outline the limitations of our work to inform future research directions. Before deploying our method in real-world applications, practices such as data association and camera pose estimation are necessary for effective multi-object mapping. Our approach relies on text descriptions and segmented images from a segmentation algorithm as additional inputs. While these can be relatively easy to acquire using off-the-shelf models like Segment Anything, they do introduce extra input requirements. Additionally, the computation is not yet real-time, primarily due to the diffusion process and NeRF rendering. The challenges posed by the diffusion process could be mitigated by employing more lightweight generative models, such as those with smaller latent spaces. For the NeRF rendering, more efficient representations, like Gaussian splatting, could improve performance. Parameters such as the number of optimization steps can be adjusted to balance efficiency and effectiveness. Our current focus is on the priors of individual objects, leading us to evaluate them independently. However, more scene-level information and priors could enhance global consistency, such as cross-object relationships, structural knowledge about objects, and scene graphs. These aspects could provide valuable avenues for future work.

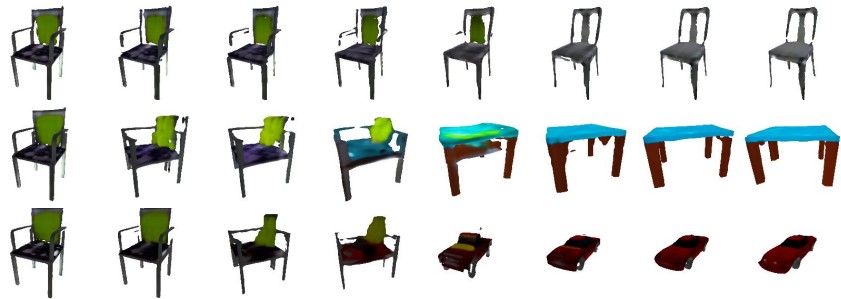

Figure 11: Latent Space Interpolation: visualizing the transition of Shap-E generated models from (1) a chair to another chair; (2) a chair to a table; (3) a chair to a car.

| Text Prompt | Generated Shapes | | | | |
|---|---|---|---|---|---|
| A chair | | | | | |
| A green chair | | | | | |

Figure 12: Text-Conditioned Generation: Shap-E can generate diverse shapes based on given text prompts. The application of more detailed text prompts presents an intriguing future direction for further constraining the shape and pose mapping process.

## 6.7 Analysis of the Generative Model Shap-E

**Latent Space Interpolation**. We illustrate a visualization of latent space interpolation from one chair to another, from a chair to a table, and from a chair to a plane in Figure 11. Unlike DeepSDF [27], which utilizes a 64-dimensional latent vector for an SDF-based shape, Shap-E employs a considerably larger latent space for a NeRF-based shape, with a dimension of $1024 \times 1024$. Despite its high dimensionality, linear interpolation still provides a meaningful transition for changes in both texture and geometry. A smooth latent space aids the optimization process when incorporating gradients from both observations and priors.

**Generation from Text Prompt**. The Shap-E model is capable of generating a variety of shapes based on a given text prompt, as demonstrated in Figure 12. The attributes specified in the text prompts, such as color, can influence the output shapes to a certain degree. The use of more complex text prompts, such as descriptions from large language models (LLMs) to assist in mapping object shapes and poses, presents an intriguing avenue for future research.

## 6.8 Derivation of Optimization with Prior

We provide the proof for Equation 5 in the main content. Given $M$ observation frames $\{F_i\}_{i=1}^M$, and a condition $C$, we aim to estimate a Maximum Likelihood Estimation for the unknown variable pose $\mathbf{T}$ and shape $\Theta$. We start from a joint distribution of $P(\mathbf{T}, \Theta | F_1, ..., F_M, C)$, and aim to get:

$$\hat{\mathbf{T}}, \hat{\Theta} = \arg \max_{\mathbf{T}, \Theta} P(\mathbf{T}, \Theta | F_1, ..., F_M, C) \tag{17}$$

According to Bayes' rule:

$$P(\mathbf{T}, \Theta | F_1, ..., F_M, C) = \frac{P(F_1, ..., F_M | \mathbf{T}, \Theta, C) P(\mathbf{T}, \Theta | C)}{P(F_1, ..., F_M | C)} \tag{18}$$

Considering that any observation frames $F_1, ..., F_M$ are independent to the prior condition $C$, and we can assume the prior of the observation $P(F_i)$ is a constant, thus, $P(F_1, ..., F_M|C)$ is a constant. We can get:

$$P(\mathbf{T}, \Theta|F_1, ..., F_M, C) \propto P(F_1, ..., F_M|\mathbf{T}, \Theta, C)P(\mathbf{T}, \Theta|C) \qquad (19)$$

Then, we consider the observation part $P(F_1, ..., F_M|\mathbf{T}, \Theta, C)$. Since the observations $F_1, ..., F_M$ are conditionally independent among each other given $\mathbf{T}$ and $\Theta$, and are independent to $C$, the likelihood can be factorized as:

$$P(F_1, ..., F_M|\mathbf{T}, \Theta, C) = \prod_i P(F_i|\mathbf{T}, \Theta) \qquad (20)$$

Since we model the pose $\mathbf{T}$ and shape $\Theta$ separately, they are independent to each other. Further considering that the condition $C$ only applies to the shape, we have:

$$P(\mathbf{T}, \Theta|C) = P(\mathbf{T}|C)P(\Theta|C) = P(\mathbf{T})P(\Theta|C) \qquad (21)$$

We assume uniform distribution for the object pose $\mathbf{T}$, so that $P(\mathbf{T})$ is a constant. So we have:

$$P(\mathbf{T}, \Theta|C) \propto P(\Theta|C) \qquad (22)$$

Inserting the observation part (Eq 20) and the prior part (Eq 22) into the joint distribution (Eq 19), we can estimate the unknown variables through:

$$\hat{\mathbf{T}}, \hat{\Theta} = \arg\max_{\mathbf{T}, \Theta} \prod_i P(F_i|\mathbf{T}, \Theta)P(\Theta|C) \qquad (23)$$

Finally, taking the logarithm, we can get a more convenient form for numerical optimization:

$$\hat{\mathbf{T}}, \hat{\Theta} = \arg\max_{\mathbf{T}, \Theta} \sum logP(F_i|\mathbf{T}, \Theta) + logP(\Theta|C) \qquad (24)$$

