# OpenReview forum: "Toward General Object-level Mapping from Sparse Views with 3D Diffusion Priors"
_robot-learning.org/CoRL/2024/Conference — CoRL 2024_

### Official Review · Reviewer_UoFm · 2024-07-20
**GOM Paper Review**

**Originality:** 3
**Technical Quality:** 4
**Clarity Of Presentation:** 3
**Potential Impact:** 3
**Recommendation:** 4
**Confidence:** 3

**Review:**

The paper is clear in it's purpose/contributions. The authors make clear how recent advancements in 3D diffusion models have opened the door for diffusion priors to be used, as well as the gap in the field for identifying both shape and pose of objects.

Their methodology provides details as to they take the diffusion priors from the generative diffusion network (originally trained for 3D object generation) and apply them to their task of shape and pose mapping.

Finally the paper's authors provide experiments comparing their proposed diffusion-incorporating method to a handful of existing ones. The results consistently validating their proposed method can be considered a strength, while the use only of Scannet data for experimental results can be considered one of the paper's weaknesses.

With regards to this conference's focus on robotics, RGBD cameras are very common, and the ability to quickly and reliably map surrounding objects in undeniably useful for a large range of robotics applications. The paper demonstrates improved reliability for shape mapping, however whether the inclusion of diffusion significantly slows down processing is unclear.

**Quality Of The Limitations Section:**

3

**Questions For Rebuttal:**

1) The paper acknowledges diffusion models are slow to train, however I am curious what is the inference time of your method compared to the other ones? In the context of image generation, VAE and GAN are "Fast" while Diffusion is slow (but accurate) in inference. If your improved results similarly come at the cost of longer processing time this should be clearly outlined as a limitation.

2) Similarly Figures 1 and 2 both imply through their use of arrows that the text prompt e.g. "A Chair" is provided not just during training but during inference.

If the proposed method requires an extra input alongside the RGBD photos (brief text description of the object) then this should be clearly outlined in the limitations section. An autonomous robot for example, would not necessarily know that the shape it is trying to identify is a chair.

**Robotics Focus:**

3

**Summary Of Paper:**

Author's propose a novel method (GOM) to identify an object's 3D shape and pose from a small subset of RGBD images.

**Summary Of Recommendation:**

Paper introduces a novel method to integrate diffusion into 3D shape mapping for RGBD, validated by results. This is relevant research for CoRL, and concerns adressed by rebuttal, so strong accept

---

### Official Review · Reviewer_3JtG · 2024-07-21
**Novel Use of 3D Diffusion Priors in Object Reconstruction**

**Originality:** 4
**Technical Quality:** 4
**Clarity Of Presentation:** 5
**Potential Impact:** 3
**Recommendation:** 4
**Confidence:** 4

**Review:**

Strengths
- The idea to combine 3D diffusion priors with time series online measurements is timely and relevant to both the robotics and computer vision communities.
- The method is clearly communicated, and the paper is well organized.
- The use of the denoising objective to update object estimates in a 3D mapping framework is novel and exciting.

Weaknesses
- For the experiments, each map appears to include only a single object (this seems to be implied in the Appendix (L7-9)). As stated in the draft, object-level mapping generally builds maps of multiple 3D objects. The present results are closer to multi-view single object-level reconstruction.
- Because only a single object is mapped, there is no discussion of other practicalities that would be required to actually deploy the proposed method, such as data association.
- The only objects considered are large-scale furniture objects, which tend to have very repeatable structures and easy to obtain semantic labels. It is unclear how this method would perform on more "open world" scenarios with more diverse objects. Even qualitative examples in these scenarios would be informative.

Other
- I suggest refraining from calling the steps where the diffusion prior is used to optimize the loss as “diffusion steps” to avoid confusion with the iterative denoising processes.
- A qualitative figure showing the 10 input images in addition to their 3D poses and the resulting shape reconstruction (rendered in 3D coordinates) would be a compelling addition.
Table 2 appears to reproduce the numbers in Table 1 for the Chair column, but Table 1 does not have the Shap-E + ICP result in the 10 view column.

**Quality Of The Limitations Section:**

1

**Questions For Rebuttal:**

- L233 notes that Shap-E + ICP “exhibits a low CD but struggles to faithfully reconstruct objects”. The CD is significantly lower than the proposed method; it would be informative to expand on why CD does not correlate to faithful reconstruction in this case.
- Are the gradient fields shown in Figure 2 actual fields (as implied by L181), or for illustration? Very interesting if actual fields, but there should be some details on how the figure is generated in 2D.
- L29 in the Appendix indicates that the “Jointly Optimize and Diffuse” paradigm “simultaneously considers” both the measurements and the prior. However, L190-196 in the main text seems to indicate that the process is iterative (i.e., the prior is applied first, then the observation). Is it important to update the prior and observation updates iteratively? Or can they be updated in a single consolidated step?
- A limitations section has not been provided; this must be addressed.

**Robotics Focus:**

3

**Summary Of Paper:**

This paper proposes a novel method of object-centric mapping that uses a 3D diffusion prior as a shape prior. Given a set of RGB-D views, an iterative optimization process is proposed that uses a diffusion prior as well as a rendering loss to estimate the object pose and shape. The diffusion prior update is approximated by the estimates of a 3D diffusion model (in this case, Shap-E). The authors present experiments on the ScanNet dataset, and extensive ablations in the Appendix.

**Summary Of Recommendation:**

Given the novelty of the idea and corresponding quantitative results, I recommend a score of SA.

---

### Official Review · Reviewer_vp6P · 2024-07-22
**Initial Review**

**Originality:** 3
**Technical Quality:** 3
**Clarity Of Presentation:** 4
**Potential Impact:** 3
**Recommendation:** 3
**Confidence:** 2

**Review:**

Strength
1. The paper is well-written and well-organized.
2. The proposed probabilistic optimization framework can effectively fuse the shape prior from a diffusion model with the RGB sensor measurements.
3. The proposed method is shown to outperform the baselines both quantitatively and qualitatively.

Weakness
1. This work relies on segmentation results to provide the category for the text prompt. Currently, it is from ground truth segmentation. The sensitivities to the quality of the segmentation method is unknown.

2. The authors use Intersection over Union (IoU) to evaluate pose accuracy. However, this metric might be affected by the quality of the reconstruction. I recommend the author also report pose errors separately.

3. One of the performance metrics of object level mapping is the consistency of the map. Currently, the different objects are evaluated separately. I think the paper can benefit from another evaluation that includes the whole scene and with results between multiple objects.

4. The limitation of the work is not well addressed in the paper.

**Quality Of The Limitations Section:**

1

**Questions For Rebuttal:**

1. Can the method perform well if only a single view is provided to the system?
2. How would the proposed system perform for out of distribution objects? (For example, if the Shape-E provides an incorrect shape prior because the input object is out of distribution.)
3. What is the runtime of the proposed system?

**Robotics Focus:**

2

**Summary Of Paper:**

This paper proposes a General Object-level Mapping system (GOM) which uses a diffusion model to generate shape prior and outputs NeRF representation for both texture and geometry of the scene. A probabilistic optimization scheme is proposed to fuse the sensor observations with the diffusion prior. The proposed method is evaluated on the ShapeNet dataset and is shown to outperform the baselines both qualitatively and quantitatively.

**Summary Of Recommendation:**

The paper proposes an interesting approach to fuse shape prior from a diffusion model with multi frame measurements. The system is shown to outperform the baselines in different metrics. Although there are some questions to be clarified, I find the proposed method a good addition to the current literature. As a result, I’m recommending a weak accept.

---

### Author Rebuttal · Authors · 2024-08-12

We appreciate the reviewers’s constructive feedback. We provide our rebuttal contents in the uploaded document. We hope our answers can help clear the concerns. Please feel free to ask if you have any further questions.

---

### Decision · Program_Chairs · 2024-09-04

**Decision:**

Accept

**Comment:**

This paper introduces a General Object-level Mapping (GOM) system that uses a diffusion model to generate shape priors and NeRF representations for scene texture and geometry. The system includes a probabilistic optimization scheme to fuse sensor observations with the diffusion prior. The method estimates object pose and shape from RGB-D views through iterative optimization, leveraging a pre-trained 3D diffusion model (Shap-E).

All reviewers acknowledge that the paper is well-written and well-organized. The idea is timely and relevant to both the robotics and computer vision communities. Concrete comparisons with baselines demonstrate the method's advancement both qualitatively and quantitatively.

Before the rebuttal, several concerns were raised: the method's sensitivity to the quality of the segmentation, the need for additional metrics such as pose errors, the consistency of the map, and the quality of data association.

During the rebuttal, the authors did an excellent job addressing these concerns by providing detailed clarifications regarding the method and evaluation details. While the AC acknowledges the contribution of the work, they also echo some reviewers' concerns about the lack of concrete results in larger-scale mapping and the absence of actual robotic applications enabled by the proposed framework. The inclusion of these additional results would further enhance the impact of this work in the robot learning community.